# Going Retro, Going Viral: Experiences and Lessons in Drug Discovery from COVID-19

**DOI:** 10.3390/molecules27123815

**Published:** 2022-06-14

**Authors:** Bing Wang, Dmitri Svetlov, Dylan Bartikofsky, Christiane E. Wobus, Irina Artsimovitch

**Affiliations:** 1Department of Microbiology, The Ohio State University, Columbus, OH 43210, USA; wang.13377@osu.edu; 2Svetlov Scientific Software, Pasadena, CA 91106, USA; svetlovscientific@protonmail.com; 3Department of Microbiology and Immunology, University of Michigan, Ann Arbor, MI 48109, USA; dylanbar@umich.edu (D.B.); cwobus@umich.edu (C.E.W.)

**Keywords:** SARS-CoV-2 RdRp, NiRAN, nucleotide analogs, fidaxomicin, rifamycins

## Abstract

The severity of the COVID-19 pandemic and the pace of its global spread have motivated researchers to opt for repurposing existing drugs against SARS-CoV-2 rather than discover or develop novel ones. For reasons of speed, throughput, and cost-effectiveness, virtual screening campaigns, relying heavily on in silico docking, have dominated published reports. A particular focus as a drug target has been the principal active site (i.e., RNA synthesis) of RNA-dependent RNA polymerase (RdRp), despite the existence of a second, and also indispensable, active site in the same enzyme. Here we report the results of our experimental interrogation of several small-molecule inhibitors, including natural products proposed to be effective by in silico studies. Notably, we find that two antibiotics in clinical use, fidaxomicin and rifabutin, inhibit RNA synthesis by SARS-CoV-2 RdRp in vitro and inhibit viral replication in cell culture. However, our mutagenesis studies contradict the binding sites predicted computationally. We discuss the implications of these and other findings for computational studies predicting the binding of ligands to large and flexible protein complexes and therefore for drug discovery or repurposing efforts utilizing such studies. Finally, we suggest several improvements on such efforts ongoing against SARS-CoV-2 and future pathogens as they arise.

## 1. Introduction

Throughout human history, viral infections have caused massive casualties worldwide; more than 30 million people have died from AIDS-related illnesses since the start of the AIDS pandemic in 1981 [1] (https://www.unaids.org; accessed on 2 June 2022), and more than six million people have succumbed to COVID-19 in just over two years [2] (https://coronavirus.jhu.edu; accessed on 2 June 2022). While SARS-CoV-2 remains the most urgent global threat, many other viruses with endemic or pandemic potential are present in the environment, making the development of antiviral drugs an ever-pressing need.

Broad-spectrum antiviral agents (BSAA) could provide a first line of defense before vaccines and virus-specific therapeutics can be deployed, and viral nucleic acid polymerases have been long viewed as promising targets for pan-viral drugs. Many nucleoside analogs (NA) have been developed against clinically important viruses, such as Ebola virus, influenza virus, hepatitis C virus (HCV), and human immunodeficiency virus (HIV) [3]. The first antiviral drug approved in 1963, 5-iodo 2′-deoxyuridine, inhibits herpesvirus DNA polymerase [3]; the first approved inhibitor of SARS-CoV-2, remdesivir [4], targets viral RNA-dependent RNA polymerase (RdRp); and the RNA synthesis inhibitor ribavirin is active against a broad spectrum of positive- and negative-sense RNA viruses [5]. RdRp is the only protein common to all RNA viruses, from a small bacteriophage Qβ, which infects *Escherichia coli*, to the large and complex coronaviruses such as SARS-CoV-2 [6]. These enzymes share several structural elements that mediate RNA synthesis via a conserved catalytic mechanism [7] yet are distinct from the multi-subunit nuclear RNA polymerases (RNAP) that carry out transcription in the host cell.

A drug’s path from initial discovery to regulatory approval commonly takes more than a decade [8]. Unprecedented public health and economic disruptions brought by the COVID-19 pandemic commanded immense government, industry, and grassroot resources toward repurposing of existing drugs on an emergency schedule. Not surprisingly, assessment of known antiviral NAs as potential inhibitors of SARS-CoV-2 RdRp [9] demonstrated that many, including molnupiravir [10], favipiravir [11], AT-527 [12], and sofosbuvir [13], inhibit viral RNA synthesis. In parallel, hundreds of virtual screening campaigns sought to identify novel non-NA COVID-19 therapeutics among the list of existing drugs, and numerous experimental screens were used to identify molecules that inhibit viral replication/enzyme activities [14]. A long list of small molecules that could bind to RdRp has been compiled, but only a few have been experimentally validated.

Studies of RNA synthesis in coronaviruses, dramatically accelerated at the onset of the pandemic, revealed several unique properties of the SARS-CoV-2 replication machinery that must be taken into consideration during drug discovery. First, a minimally active RdRp is composed of four non-structural proteins (nsps), the catalytic nsp12 and accessory nsp7 and nsp8 proteins, the latter present in two copies [15,16]. Second, SARS-CoV-2 RdRp is much faster and more error prone than other well-studied viral enzymes [11,17]. Third, RdRp is a bifunctional enzyme: in addition to the C-terminal “right hand” domain that harbors the conserved RdRp active site, nsp12 also contains an N-terminal Nidovirus RdRp-Associated Nucleotidyl transferase (NiRAN) domain, which modifies viral RNA and proteins [18,19,20,21,22]. Both nsp12 active sites are essential for viral growth [19] and targeting either site is expected to inhibit SARS-CoV-2. Fourth, computational and functional studies suggest that, like multi-subunit bacterial RNAPs, SARS-CoV-2 RdRp may be controlled allosterically [17,23] and several hypothetical allosteric pockets have been identified in nsp12 [24,25,26]. Finally, the transcribing RdRp associates with other nsps to form a large replication–transcription complex [16], which presents multiple hypothetical targets for small-molecule inhibitors. NAs that bind to the RdRp and NiRAN active sites could inhibit catalysis or serve as alternative substrates, generating aberrant adducts (Figure 1). Molecules that bind to other sites could sterically block RdRp interactions with RNA or with other nsps (7, 8, 9, 13, etc.), whereas ligands that bind to allosteric sites could inhibit catalysis in either active site. Non-nucleoside allosteric inhibitors have been used as antivirals against HIV and HCV infections [3], and most antibiotics that target (unrelated) bacterial RNAPs act allosterically [27]. Allosteric inhibitors may be less susceptible to emerging resistance and are valuable additions to combination therapies [28].

In this work, we used our established in vitro RNA assays of both catalytic activities of SARS-CoV-2 RdRp to evaluate the efficacy of hypothetical small-molecule inhibitors, including several natural products, identified in silico. Our results support a notion that in silico docking approaches have limited utility in predicting binding sites/poses of ligands targeting large and flexible proteins, even when performed in a dynamic mode [29]. We found that tobramycin and rutin, docked in the active site of the NiRAN domain [30], had no effect on NMPylation, whereas two clinically used antibiotics that were docked to the NiRAN/RdRp interface [26], fidaxomicin and rifabutin, inhibited RNA synthesis by SARS-CoV-2 RdRp in vitro and viral replication in cell culture. However, our results suggest that these antibiotics likely bind neither to the RdRp active site nor to a hotspot at the NiRAN/RdRp interface proposed to bind these and diverse other predicted inhibitors of RdRp [26]. Our findings and published reports [31,32,33] suggest that there are significant limitations to many common approaches to computational drug (re)discovery and that these and other natural products deserve a second look in the search for antiviral drugs.

## 2. Results

### 2.1. SARS-CoV-2 NiRAN Domain as a Target for Small-Molecule Ligands

Identification of an additional enzymatic activity within the catalytic subunit of RdRp [34] raised a possibility that inhibitors targeting the NiRAN domain, including many existing NAs, could be added to the arsenal of antiviral drugs for SARS-CoV-2. Although NiRAN has no homologs in other RNA viruses, it is present in all nidoviruses and is essential for viral replication [34]. The NiRAN domain is structurally homologous to AMPylases [35,36], which catalyze AMP transfer to target proteins [37], and transfers NMPs to several nsps [19,20,21,38]. In contrast to AMPylases, which transfer AMP to hydroxyl groups of Ser, Thr, and Tyr residues to generate stable modifications [37], the NiRAN active site (AS2 in Figure 1) targets primary amines to yield labile adducts [19], which are used as transient catalytic intermediates in other enzymatic systems [39].

In SARS-CoV-2, the Asn1 residue of nsp9, an essential RNA-binding protein, is a target for NiRAN, and substitutions of nsp9 Asn1 and the catalytic residues in AS2 block virus propagation in cell culture [19,34]. NiRAN has also been reported to mediate viral RNA capping, either directly [18] or via an RNA-nsp9 intermediate [22]. Small molecules could interfere with NiRAN activity by binding to nsp9, AS2, or an allosteric site. A natural product inhibitor that binds to nsp9 has been recently identified [40], and structures with ligands bound to AS2 [12,18,35,41,42] could guide design of NiRAN inhibitors. An in silico docking study identified several putative NiRAN ligands among FDA-approved protease and kinase inhibitors as well as antibiotics [30].

Both AS1 and AS2 bind NTPs and catalyze NMP transfer, and recent successes in using NA antivirals against COVID-19 support the notion of dual-target NA antivirals. FDA-approved NA antivirals are incorporated into the nascent RNA and exert their effects either as chain terminators or as mutagens [16,43]. AS2 has been shown to transfer all natural NMPs, and some analogs, to target proteins [20,21], implying that mis-NMPylation may happen, possibly with deleterious consequences for viral replication. To act as an inhibitor, an NA must outcompete natural NTPs; indeed, remdesivir triphosphate (RDV-TP) outcompetes ATP in different in vitro RNA synthesis assays [44,45,46] and AT-527 triphosphate (a.k.a. AT-9010), a 2′-fluoro-2′-C-methyl GTP, has been reported to be an efficient competitor of GTP in both active sites [12].

While base-pairing is essential for NA accommodation in AS1, no base-specific contacts have been observed in several structures of NiRAN-bound nucleotides [18,35], and natural NTPs appear to bind to AS2 similarly during nsp9 NMPylation [20]. However, preferential binding of GTP is required for RNA capping [22], and recent reports show that NAs may bind to AS2 in strikingly different poses [12,42]: in some transcription complexes, GTP [42] and its analog AT-527 diphosphate [12] bind to AS2 in a flipped orientation (Figure 2A) with the base located in a guanine-specific pocket that is absent in the apo-NIRAN structure [42]. These observations suggest that AS2 has considerable plasticity, which may be a shared feature of AMPylases: *E. coli* Doc, a toxin from the Fic family of AMPylases, transfers the ATP γ-phosphate (rather than AMP) to a Thr residue of the translation elongation factor EF-Tu, blocking binding of aminoacylated tRNAs [47].

Knowledge of the effects of NAs on both SARS-CoV-2 RdRp activities is essential for their evaluation as dual-target inhibitors. Many groups have studied NA incorporation into RNA, but the effects of NAs on NiRAN activity have not been systematically analyzed. This knowledge is important not only for antiviral repurposing of NAs but also for our understanding of the molecular mechanism of NiRAN catalysis and its regulation. AT-527-TP and a clinically relevant UTP analog carrying the same sugar modification, sofosbuvir triphosphate (SOF-TP; Figure 2B), have been reported to outcompete native NTPs in AS2 during the NMPylation reaction [12]. The altered, flipped orientation of AT-527-TP (Figure 2A) could explain its preferential binding to AS2 [12], but whether SOF-TP also binds to AS2 in a flipped orientation remains to be determined. While our observations that remdesivir monophosphate is transferred to nsp9 [20] imply that RDV-TP binds to AS2 in the same orientation as its homolog ATP [35], we found that RDV-TP is a poor NiRAN substrate and competitor, and modeling suggests that the 1′-cyano group of RDV-TP may clash with His75 in AS2 (Figure 2A).

Comparison of published data on NMPylation by NiRAN is complicated by the use of different experimental conditions; most importantly, the identity and concentration of the catalytic metal ion. In our experiments, we use Mg^2+^ at physiological concentrations, when efficient NMP transfer is observed to nsp9 but not to nonspecific protein substrates, such as BSA [20]. Here, we asked whether GTP and UTP analogs carrying 2′ substitutions have different effects on nsp9 modification. We assayed [α^32^P]-GMP transfer to nsp9 in the presence of different cold NTP competitors (Figure 2C). As expected, GTP reduced NMPylation two-fold when present at equimolar concentration (25 μM; Figure 2D); the same effect was observed with SOF-TP and 2′-azido-GTP and -UTP. Consistent with our previous observations [20], RDV-TP did not compete efficiently, whereas AT-527-TP was a better competitor, with 50% inhibition observed at 12.5 μM (Figure 2D). In contrast to an earlier study [12], which concluded that AT-527-TP and, to a lesser extent, SOF-TP were able to outcompete natural NTPs in AS2, we did not observe any differences between GTP and SOF-TP, and the AT-527-TP effect was modest. A possible explanation for this discrepancy is that AT-527-TP (and possibly also SOF-TP) may be hydrolyzed to its diphosphate form by AS2, as shown in [12]; in turn, we found that AT-527-DP and SOF-DP are poor competitors of GTP in the AS2, as they are unable to efficiently inhibit GMP transfer to nsp9 (Figure 3B). However, we think that experimental conditions, particularly the use of Mg^2+^ in our study vs. Mn^2+^ by Shannon et al. [12], are more likely responsible for the observed differences; we note that the AT-527-TP/NiRAN structure in Figure 2A was obtained in the presence of Mg^2+^.

Our results suggest that many NAs will be similarly effective in competing with native NTPs during NMPylation. The presence of additional groups may affect NA interactions with AS2 either positively or negatively, as observed with AT-527-TP and RDV-TP, respectively, and it may be possible to design novel NAs that will bind to NiRAN with high affinity. There are no data to suggest, however, that the same NAs will be effective against both active sites; indeed, RDV-TP is the best known NTP competitor during RNA chain synthesis [44,45,46] but a very poor substrate for NiRAN (Figure 2D). On the other hand, 2′-azido NTPs and SOF-TP are as efficient NMPylation competitors as their cognate NTPs (Figure 2D), but are very easily outcompeted by NTPs during RNA chain synthesis [45,46], whereas AT-527-TP is able to compete for binding to both sites [12]. 

While the SARS-CoV-2 RdRp AS1 can bind NAs that contain sugar and base modifications, these ligands must be structurally similar to natural NTPs to be efficiently incorporated into the nascent RNA. By contrast, AS2 may be able to accommodate diverse ligands, including nucleotide di- and tetra-phosphates. Pyrophosphate PP_i_ binds to NiRAN [41] and reverses NMPylation [20] suggest that different phosphorylated molecules can be explored as potential NiRAN inhibitors; e.g., the antiviral drug foscarnet modestly inhibits NMPylation [20], and thus testing other stable PP_i_ analogs could be worthwhile.

Structural studies reveal that the ligand-free NiRAN domain is largely disordered [36] and AS2 remains relatively open even in the presence of a bound ligand in some contexts [35,41], suggesting that molecules that do not resemble nucleotides or PP_i_ may also be able to bind to NiRAN. Consistently, an in silico docking study identified several potential NiRAN inhibitors among well-known FDA-approved and experimental therapeutics, including antibiotics, anti-inflammatory agents, inhibitors of kinases and proteases, and dietary supplements [30]. Some of these molecules, including tobramycin (an aminoglucoside) and rutin (a flavonoid antioxidant), were predicted to bind to NiRAN with affinities comparable to native NTPs, making direct contacts with AS2 residues implicated in catalysis (e.g., Arg116, Asp208, and Asp218; Figure 3A). Since tobramycin and rutin are commercially available, we tested their effects on NMPylation. We found that neither compound inhibited GMP transfer to nsp9, even when present at 0.4 mM (Figure 3B).

Our results confirm that AS2 is promiscuous and readily accommodates diverse NAs, including those that are poor substrates for the RdRp AS1, such as SOF-TP and 2′-azido NTPs. Nucleotides are expected to readily exchange in AS2, and an efficient NA inhibitor would need to either bind to NiRAN very tightly, perhaps covalently, or to trigger “lethal NMPylation” through mis-incorporation into viral RNA and proteins. We also show that in silico docking of small molecules into NiRAN may be misleading, which could be expected with such a dynamic target. Importantly, however, the NMPylation reaction should be adaptable to high-throughput screening, e.g., with fluorescent substrate analogs [40].

In addition to NMPylation, AS2 has been shown to mediate a mechanistically distinct reaction wherein RNA is transferred to nsp9 to initiate the RNA capping pathway [22]. Effects of NAs on viral RNA capping and the roles of both reactions in the SARS-CoV-2 lifecycle remain to be investigated.

### 2.2. Antibiotics Targeting Bacterial RNA Polymerase Also Inhibit SARS-CoV-2 RdRp

While NAs predominate among the viral nucleic-acid synthesis inhibitors [3], most antibiotics that target bacterial RNAPs are not substrate analogs. Rifamycins (RIFs) and fidaxomicin (FDX), small molecules discovered as natural products, are used to treat infections caused by *Mycobacterium tuberculosis* [48] and *Clostridium difficile* [49], respectively. RIFs and FDX bind to different sites on bacterial RNAP, distant from the active site, and target different steps of RNA synthesis: FDX inhibits formation of productive transcription initiation complexes, whereas RIFs block the nascent RNA chain extension, sequestering the enzyme at promoters in an abortive cycle of synthesis and release of short RNAs [27]. Interestingly, these inhibitors also have anti-viral effects. Rifampin inhibition of vaccinia virus morphogenesis [50] was discovered contemporaneously with its antibacterial activity [51]; subsequent structural studies demonstrated that rifampin blocks interactions between viral scaffolding proteins [52]. FDX was identified in silico as a potential ligand for the RdRp domain of Zika virus NS5 protein [32] and demonstrated to directly bind to RdRp and inhibit replication of Zika and a related Dengue virus [32]. 

An in silico docking report suggested that RIFs, FDX, and other small molecules including a prodrug form of RDV-TP and ivermectin, may bind to SARS-CoV-2 nsp12 [26] at a site that could be involved in allosteric signaling [24]. This hypothetical hotspot lies at the interface between the NiRAN and RdRp domains (Figure 4A), and our results support the existence of allosteric communications between the two nsp12 active sites [23]. In addition to the vast untapped potential of natural products as antivirals, small-molecule inhibitors have been invaluable in elucidating molecular mechanisms of bacterial transcription, prompting us to test if FDX and RIFs modulate SARS-CoV-2 RdRp activity.

We first used an established in vitro RNA extension assay [23] to test if FDX and RIFs rifapentine and rifabutin (Figure 4B) inhibit RNA synthesis. As a control, we used suramin (SUR), a potent inhibitor of SARS-CoV-2 RdRp that blocks its binding to RNA [53]. As expected, SUR strongly inhibited RNA synthesis, with an IC_50_ of ~2 μM (Figure 4C). We found that RFB and FDX also inhibited RNA synthesis, but only at high concentrations (~100 and 50 μM IC_50_, respectively), whereas rifapentine (RPN) was even less effective (Figure 4C). Myxopyronin, a bacterial RNAP inhibitor docked onto nsp12 in another study alongside RIFs [54], did not shown any inhibition (data not shown). Small molecule inhibitors are poorly soluble in water and, in our experiments, the solvent (DMSO) was used as a control. We found that, in contrast to *E. coli* RNAP that tolerates 20% DMSO, RNA synthesis by SARS-CoV-2 RdRp is inhibited by DMSO above 2%, complicating evaluation of drug candidates.

We next tested whether RNA binding is inhibited by these natural products using a gel shift assay with the same RNA scaffold as employed for the RNA extension assay (Figure 4D). As shown previously [53], SUR prevented the formation of RdRp•RNA complex. By contrast, neither RFB nor FDX had any effect.

Since RFB and FDX are predicted to bind at the RdRp/NiRAN interface, it is possible that they may alter the function of the NiRAN active site. However, neither FDX nor RFB inhibited transfer of the ^32^P-GMP to nsp9 (Figure 4E). The lack of the effect of SUR is consistent with the location of its binding sites on nsp12 and the apparent lack of conformational changes in RdRp•SUR complex [53].

To test the antiviral activity of FDX and RFB in a cell-based assay, we infected Vero E6 cells with SARS-CoV-2 WA1 strain in the presence of increasing concentrations of FDX or RFB. We found that both compounds completely inhibited SARS-CoV-2 infectious virus production at the highest concentration tested (Figure 4F), which for each compound maintained cell viability above 80% (data not shown). This demonstrates the antiviral efficacy of both compounds, although RFB had greater efficacy, reaching near-complete inhibition at 40 μM vs. 200 μM required for FDX. Consistent with these findings, a recent cellular reporter assay demonstrated inhibition of SARS-CoV-2 RdRp by rifampicin [33].

RIFs and FDX are natural products identified as inhibitors of bacterial RNA synthesis, so their modest effects on SARS-CoV-2 RdRp are perhaps not surprising, but synthetic chemistry approaches have been used to modify both classes of molecules [55,56,57]. In the absence of an experimental structure of SARS-CoV-2 RdRp bound to either FDX or RIFs, we decided to use mutagenesis to probe their contact sites identified in silico [26]. Although we were concerned that very diverse molecules were predicted to bind to the same site at the NiRAN-RdRp interface (Figure 5A), inhibitors of bacterial RNAP are known to exert their effects by binding to interdomain surfaces [58]. We selected six residues modeled to make contacts to both FDX and RFB, Tyr32, Tyr129, Asn138, and Thr141 in the NiRAN domain and Ser709 and Asn781 in RdRp (Figure 5A). Since both ligands are large and could make many contacts, we used non-conservative replacements that alter side-chain size to maximize potential effects. We found that all mutant nsp12s retained RNA synthesis activity (Figure 5B), although reduced relative to the wild-type enzyme; the N138W RdRp displayed the largest defect. We then tested the mutant enzymes’ response to intermediate concentrations of RFB and FDX selected to observe changes in either direction; we used SUR, which binds to two sites distinct from the hotspot [53] as a control. We found that none of the selected nsp12 substitutions conferred resistance to RFB or FDX, and N138W was hypersensitive, in line with the general loss of activity in this mutant enzyme. Therefore, we conclude that the binding site/pose identified by in silico docking is incorrect.

## 3. Discussion

Joining many colleagues, in 2020 we initiated mechanistic studies of the SARS-CoV-2 RNA synthesis and embarked on a quick search for non-NA drugs that could be deployed against COVID-19. Lacking expertise in drug discovery, we decided to test several hypothetical inhibitors of RdRp identified in silico by others and readily available to us. We quickly found that two commonly used antibiotics targeting bacterial RNAP, FDX, and RFB, inhibited RNA synthesis by SARS-CoV-2 RdRp, albeit weakly. However, our failed attempts to validate their hypothetical binding site [26] made us realize the challenges that arise during the docking of small molecules into dynamic protein complexes and, by extension, drug discovery or repurposing campaigns utilizing docking against such targets.

The COVID-19 pandemic has prompted hundreds, if not thousands, of such virtual screening campaigns aimed at repurposing FDA-approved drugs and other commonly used molecules (e.g., food supplements) as antivirals [14]. Repurposing is sometimes viewed as an inferior short-cut to drug development, unable or unlikely to deliver the “best in class” molecule [59]. However, these calculations change during a pandemic when even modest improvements in treatment options could save thousands of lives. Based on existing clinical safety data, FDA-approved drugs can bypass the first phase of clinical trials, reducing costs and time to deployment.

Antiviral drugs can be categorized into two kinds: those that act directly against the pathogen, by targeting some essential component of its viral machinery (i.e., direct acting antivirals), versus those that act indirectly, by targeting some component of the host’s cellular machinery required for infection (i.e., host-based antivirals). Approaches to drug discovery, whether the identification of novel therapeutics or the repurposing of existing ones for different or additional contexts, can similarly be divided into two types. The first seeks to determine a druggable target, whether an antigen or a molecule in the infected organism with which an antigen interacts; the second attempts, given a known or putative target, to find a suitable drug against it. The criteria for such suitability obviously include efficacy and safety. Here, focusing on direct acting antivirals, we discuss each kind of approach in the light of findings by us and others, in a “bad news first” fashion, concluding with positive reassessments and recommendations. 

### 3.1. Identification of Druggable Targets

Among the components of SARS-CoV-2 itself, the focus has been predominantly on RdRp and the main 3CL protease (M^pro^): these enzymes are indispensable for the SARS-CoV-2 replicative cycle and most viral inhibitors target either polymerases or proteases, making for straightforward repurposing and a greater likelihood of discovering a “best in class” inhibitor. The RNA synthesis active site (AS1) within RdRp is particularly attractive, because the chemistry of the nucleotide addition reaction involved is well-understood and the active sites are conserved across RNA viruses, yet different from cellular enzymes [7]. Furthermore, many NAs already exist, some of which also inhibit reverse transcriptases [3,60]. Finally, given that RdRp is the only protein present in all RNA viruses, there exists the tantalizing possibility that a BSAA might be found or created within the chemical space of NAs. Therefore, NAs have quite understandably risen to the forefront of drug discovery efforts against SARS-CoV-2 [60]. The presence of a second essential NTP-dependent active site (AS2) in the NiRAN domain raised hopes that a “dual-target” inhibitor that effectively competes with natural nucleotides in both active sites can be found. 

The bad news is that, from our perspective, no therapeutically suitable BSAAs effective against RdRp of all RNA viruses are likely to exist. Although the RdRp AS1 geometry is highly conserved across RNA viruses, the remainder of the enzyme is quite diverse. Consequently, any inhibitor binding outside of the active site is unlikely to be effective beyond the most closely related viruses. That only leaves candidates that bind to AS1, and these will almost exclusively be NAs, given the highly stringent molecular recognition of cognate NTPs required for an RdRp to quickly and processively transcribe the viral genome.

Although NA triphosphates are certainly effective against viral enzymes, they can also inhibit cellular and mitochondrial polymerases, since they all perform catalysis in fundamentally similar ways [61]. The mechanism by which a given NA inhibits a particular polymerase depends on whether the aberrant RNA can be extended after incorporation; whether the misincorporated NA is recognized by the enzyme as an error; and whether a proofreading mechanism that can correct the error exists. While some NAs trigger immediate termination, others do not block RNA synthesis after their (monophosphate) incorporation, and their presence in the product RNA causes lethal mutagenesis during subsequent replication or transcription of the product [16]. Similar to multi-subunit cellular RNAPs, and unlike most viral RdRps, the SARS-CoV-2 enzyme uses a proofreading mechanism, which depends on ExoN, to correct errors in RNA synthesis [62]. However, not all incorporated NAs are seen as erroneous, and some were reported to be resistant to ExoN [12]. These stealthy NAs will be most successful against the virus but could also permanently mutate host genomes and/or lead to teratogenicity or embryotoxicity [63]. Both nuclear RNAPII and mitochondrial RNAP readily incorporate diverse NAs, yet while RNAPII can proofread some (but not all) errors, the mitochondrial RNAP lacks proofreading capabilities, explaining why cellular toxicity of NAs correlates with their utilization by human mitochondrial enzyme [64].

Such concerns have recently drawn intense interest due to FDA emergency use authorization of molnupiravir for treatment of COVID-19 [63]. Molnupiravir monophosphate is readily incorporated by RdRps, causing lethal mutagenesis, and is thus effective against diverse viruses (see [65] and references therein). However, molnupiravir is also a good substrate for other enzymes: its active principle, β-d-N4-hydroxycytidine (NHC), causes transition mutations [66], and its ribonucleotide form, rNHC, can mutate host genomes as well as viral ones [67]. We agree with Waters et al. [63] that the case of molnupiravir is complicated: even if the available evidence regarding its short-term safety for patients were unequivocally reassuring, the possibility of adverse long-term consequences cannot be precluded, as shown by adverse effects of retroviral inhibitors that continue to emerge even after decades of use in clinical practice [3]. At a minimum, when evaluating any NA as a therapeutic candidate, empirical verification must be obtained that the NA itself, and all its metabolic products, is either not incorporated or is subsequently excised by proofreading activity associated with each relevant host polymerase. It is important to note that even if a particular NA is ineffective or unsuitable as a drug, it could be a valuable tool for the analysis of the RdRp and NiRAN mechanisms.

The good news is that more limited BSAAs may well exist. One obvious possibility is that inhibitors of AS2 should prove effective against most, if not all, nidoviruses. We showed that NMPylation is inhibited by bisphosphonates [20], one of which (foscarnet) is an FDA-approved antiviral [3], and by diverse NAs. Far greater degree of substrate promiscuity (as compared to AS1) and the higher solvent accessibility of AS2 suggest that the NiRAN is also a worthwhile, and perhaps preferable, therapeutic target. Given the off-target effects discussed above, we view it as a positive that the best inhibitors of AS1, e.g., RDV-TP, are the worst competitors for AS2. Rather than dashing hopes of dual-target inhibitors, this fact raises the possibility of highly effective and specific inhibitors. 

### 3.2. Identification of Target-Specific Inhibitors

Once a druggable target has been identified, candidate inhibitors must be identified and prioritized for the in vitro and in vivo validation of safety and efficacy that are prerequisites for regulatory approval. Given the immense chemical space of possible small molecules, a common first step in this triaging process is the high-throughput virtual screening of libraries of such compounds in a process called molecular docking.

Docking methodologies treat the question of receptor–ligand binding as fundamentally thermodynamic in nature. Therefore, they first seek the most energetically favorable binding location and orientation (“pose”) for each ligand on a given receptor, and they then rank (“score”) the ligands in the order of their affinities of binding thus predicted [68]. Since exact calculations of these free energies are computationally intractable, researchers have always utilized approximations and constraints, some of which have increasingly been relaxed as hardware has accelerated and methods were augmented. For example, while the earliest docking studies invariably treated both ligands and receptors as rigid bodies, flexible modeling of ligands has become routine [29], and more recent techniques, such as the generation of conformational ensembles via molecular dynamics simulation, have introduced flexibility for receptors as well. Other methodological improvements have tackled particularly challenging questions, such as entropic contributions to free energies of binding and the effects of solvent structure and ligand solvation [29,68]. Within the last decade, qualitatively different advances have been achieved using artificial intelligence [59], for instance deep- and machine-learning algorithms have generated drug–target interaction network graphs [69] that successfully predicted the effectiveness of dexamethasone in reducing fatality of COVID-19 [70].

The bad news is that exceptionally dynamic proteins, such as SARS-CoV-2 RdRp, present challenges for the very axioms of in silico studies. Obviously, both AS1 and AS2 could be used to interrogate potential ligands, but neither active site is fully formed in the absence of a substrate: e.g., AS1 lacks the catalytic Mg^2+^ ion and AS2 can bind substrates in different orientations [12,35,42]. Thus, the choice of a receptor structure is not obvious even when targeting either one of SARS-CoV-2 nsp12 active sites, and many other potential targets exist in the multi-subunit transcription-replication complex.

The good news is that despite these difficulties, in silico screening campaigns have borne fruit: many small-molecule inhibitors whose candidacies they have predicted have been experimentally validated as effective against SARS-CoV-2 and COVID-19. This is true even when, as has often been the case, the predicted binding site was incorrect. Our exploration of FDX and RFB as potential RdRp inhibitors was prompted by a study in which these (and several other) molecules were docked in a site on nsp12 located far from the regions that contact RNA and nsp7 and nsp8 subunits, mid-way between the two active sites [26]. A small molecule bound to this hotspot could interrupt in allosteric crosstalk between the active sites and may alter catalysis in either site. The original docking study did not allow either ligand or receptor flexibility, which may well have altered the optimal poses it therefore predicted, and thus we were not surprised to find that substitutions of the hotspot residues did not confer resistance to either inhibitor, even when large side chains were introduced to sterically block their binding (Figure 5). Thus, our discovery that FDX and RFB inhibit SARS-CoV-2 RdRp can be considered serendipitous.

Another recent example of a SARS-CoV-2 inhibitor “mis-discovered” based on virtual screening of the DrugBank library is cobicistat. Identified as a potential inhibitor of M^pro^ by docking, cobicistat was found to inhibit viral replication in cell culture, yet it did not inhibit the protease activity [71]. Post facto molecular dynamics provided a likely explanation for the mismatch, arguing that a more thorough initial docking analysis would have not selected cobicistat as an M^pro^ ligand. Thus, although more sophisticated docking protocols can in principle improve the target-ligand prediction, in this case, the effectiveness of cobicistat on SARS-CoV-2 would have been missed.

How and why, then, are these in silico studies stumbling upon successful candidates? Part of the reason is that many drugs bind to non-specific targets, which may be quite distinct from their primary targets. This promiscuity is likely driven by commonalities of molecular shapes even among very divergent proteins. SUR, an antiparasitic drug, was also shown to inhibit entry or replication of diverse viruses through contacts to viral proteins [72,73]. SUR binds to two separate sites in RdRps from SARS-CoV-2 and human norovirus but, even though RdRp is the most conserved viral protein, the SUR-binding sites on these enzymes are not conserved [53,74]. FDX and RIFs bind to distinct sites on bacterial RNAP [27] but also bind to viral proteins to interfere with the viral life cycle in cells [32,52]. We observed RdRp inhibition by FDX and RFB in a minimal in vitro RNA synthesis assay (Figure 4C), implying that these molecules exert their effects through direct contacts to RdRp. Presumably, the same effects underlie inhibition of SARS-CoV-2 replication by FDX and RFB (Figure 4F) and by rifampin [33] in cell culture, but this remains to be determined. At present, we have no reason to think that FDX and RFB bind to the same site, or even the same subunit, of RdRp, and identification of their binding sites and the mechanisms of action would necessitate targeted analysis in the future, which we are pursuing.

These examples suggest that, since the list of FDA-approved drugs is just over 4000 compounds [75], antiviral or biochemical screens of physical drug stockpiles might have been wiser than in silico ones in this case. Even when a high-throughput assay is lacking, a low-throughput screen can be fruitful, as illustrated by a study of SARS-CoV-2 proofreading exonuclease ExoN, in which in vitro analysis of just 19 candidates identified three micromolar inhibitors, among which ebselen, a promiscuous protein binder and a known inhibitor of the main CoV protease, was the most potent, with an IC_50_ of ~3 μM [76].

### 3.3. Conclusions and Perspectives

First, what is the impact of the results we are reporting here? While we have shown that FDX and RFB are poor inhibitors in vitro and in cell culture (Figure 4), these scaffolds represent a potential path to effective antivirals. A long drought in antibacterial discovery together with a rapid rise in resistance stimulated improvements of existing antibiotics, and FDX and RIFs have been chemically modified to improve/expand their antibacterial activities and to amend undesirable features, such as low solubility of FDX [55] and activation of human pregnane X receptor by RIFs [57]. These improvements were guided by high-resolution structures of complexes with their target, bacterial RNAP [77,78,79,80], and similar information would be required to improve antiviral properties of FDX and RFB. Once the binding site is identified, a systematic medicinal chemistry campaign could lead to identification of inhibitors with higher potency and increased water solubility, a serious hurdle with SARS-CoV-2 RdRp (see Results). RIFs are particularly attractive lead molecules—they have been used as front-line drugs against tuberculosis for decades despite two shortcomings, accumulation of resistance mutations in the target *rpoB* gene and off-target effects in human cells [81]. To address these issues, hundreds of RIF derivatives have been synthesized [56,57] to increase their potency against bacterial RNAPs (including the common resistant mutants) while reducing unwanted side effects. While the latter efforts would benefit development of safe antivirals, the specific chemical modifications that improve RIF binding to bacterial RNAPs are extremely unlikely to do the same for RdRp; suitable “viral” analogs of such modifications would need to be discovered.

Second, what have we learned that is more broadly applicable, beyond these inhibitors and SARS-CoV-2 RdRp per se? While findings by us and others underscore the limitations of in silico approaches to identify inhibitors of this enzyme, progress is being made in this and other directions. Without a doubt, advances in docking and machine learning will lead to improved identification of potential leads, but the task is likely to remain daunting with highly dynamic enzymes which undergo conformational changes upon binding to substrates and other small molecules or to accessory proteins. This is because the entire approach of docking, even as it is being generalized to the “dynamic” case, does not really answer the fundamental question: what is the physiologically relevant effect of binding of a particular ligand to a particular receptor? For orthosteric ligands that bind to an active site the relevance is clear, but for allosteric ligands the effect cannot be predicted from a simple free energy of binding [82]. The available data argue that SARS-CoV-2 RdRp [23,24,42] and ExoN [83] are subject to allosteric control. 

Furthermore, while virtual screening is necessary to evaluate millions of compounds as potential novel antivirals, its value for drug repurposing now appears questionable. Just the few examples listed here support a notion that traditional, activity-based screening may identify promising inhibitors of SARS-CoV-2, and other pathogens, among the known drugs that underwent safety assessment in clinical trials. Since the space of FDA-approved drugs is so small, low- and medium-throughput approaches are tractable. We therefore advocate a paradigm shift in the conduct of such repurposing campaigns. First, the rapid development and standardization of simple and inexpensive in vitro and in vivo assays should be prioritized. Standardization is particularly important, since many relevant observables, e.g., IC_50_ data, are dependent on the protocol of the assay [84], as has been shown by the inconsistencies of COVID-19 cell-based study findings across different cell lines [14], compounded by differences among cell types. Second, libraries of all FDA-approved drugs should be assembled and maintained as a physical DrugBank to facilitate investigations of their candidacies against any new pathogens. While amassing such stockpiles is prohibitively expensive for a single academic lab, it is entirely feasible for major research universities and institutes and public health agencies. If such drug banks were stored and distributed in a cooperative fashion, this would not only support logistical resilience but also allow for the involvement of greater numbers of researchers, increasing the likelihood of reproducible results and reducing the likelihood that a promising candidate is inadvertently rejected due to an experimental error or oversight.

These considerations apply to the “known target” kind of drug discovery. What improvements can be made to the “unknown target” kind? We are heartened by the development of knowledge-driven approaches to target identification, such as the AI-assisted construction and analysis of drug–target interaction networks discussed above, and we hope that researchers from all fields will contribute to the continued advancement of these methodologies. Indeed, the NiRAN domain is a perfect illustration of the usefulness of such work: its existence and indispensability were initially hypothesized as a result of bioinformatic analysis [34] and then validated by biochemical experiments [19]; only belatedly have researchers begun to appreciate its suitability as a target for antivirals. Greater integration of our understanding of underlying physiological processes, e.g., viral entry into host cells and subsequent immune responses, into drug-discovery pipelines could prove revolutionary, particularly for the identification of host susceptibilities and suitable shields to protect them. It is important to appreciate the profound variability of human physiology; to give just one pertinent example, mitochondrial toxicity may be selectively associated with some genotypes, and consequently drugs that are toxic to some could be much safer for others [64].

The COVID-19 pandemic has produced many new drug “discoverers”; perhaps most of them, including ourselves, never envisioned undertaking such exploration. Certainly, few can make significant contributions to the development of computational chemistry methods involved in classical in silico drug discovery. However, scientists from many fields—virology, enzymology, and immunology, to name but a few—can lend their knowledge to this revolution. We encourage our colleagues across all these disciplines to do so, drawing on these experiences and lessons to fundamentally rethink how we approach drug discovery, for all existing diseases—and the next pandemic.

## 4. Materials and Methods

### 4.1. Expression Vectors and Protein Purification

Plasmids used in this study are listed in Appendix A. Expression vectors for wild-type SARS-CoV-2 nsp7/8/9/12 proteins were described previously [24], and the nsp12 mutant plasmids were constructed by standard molecular biology approaches with restriction and modification enzymes from New England Biolabs (Ipswich, MA, USA), taking advantage of the existing or silent restriction sites engineered into the nsp12 coding sequence. DNA oligonucleotides for vector construction and sequencing were obtained from Millipore Sigma (Burlington, VT, USA). Sequences of all plasmids were confirmed by Sanger sequencing at the Genomics Shared Resource Facility (The Ohio State University) and will be available upon request.

The expression and purification of nsp7/8/9 were performed as described previously [24]. Nsp12 variants were overexpressed in *E. coli* BL21 (DE3) cells (Novagen, Darmstadt, Germany, Cat#69450). Cells were cultured in terrific broth (Research Products International (RPI), Mount Prospect, IL, USA, Cat#T15000) at 37 °C to an OD_600_ of ~0.6 and the temperature was lowered to 16 °C. Expression was induced with 0.1 mM isopropyl-1-thio-β-D-galactopyranoside (IPTG; Goldbio, St. Louis, MO, USA, Cat#I2481C25) overnight. Induced cells were harvested by centrifugation at 8000× *g* for 10 min at 4 °C and resuspended in 50 mM HEPES, pH 7.5, 300 mM KCl, 5% glycerol, 2 mM MgCl_2_, 1 mM phenylmethylsulfonyl fluoride (PMSF; ACROS Organics, Geel, Belgium, Cas#329-98-6), 10 mM imidazole, 10 mM β-ME, and lysed by sonication. The cleared lysate was applied to Ni^2+^-NTA resin (Cytiva), washed with 50 mM HEPES, pH 7.5, 300 mM KCl, 5% glycerol, 2 mM MgCl_2_, 10 mM β-ME, 0.1 mM PMSF, 50 mM imidazole, and eluted with 50 mM HEPES, pH 7.5, 50 mM KCl, 5% glycerol, 2 mM MgCl_2_, 10 mM β-ME, 0.1 mM PMSF, 500 mM imidazole. The eluted protein was further purified by Resource Q (Cytiva, Marlborough, CT, USA, Cat#17117701) with linear elution between Q-buffer A (50 mM HEPES, pH 7.5, 5% glycerol, 2 mM MgCl_2_, 10 mM β-ME) and Q-buffer B (50 mM HEPES, pH 7.5, 1 M KCl, 5% glycerol, 2 mM MgCl_2_, 10 mM β-ME). Then the fusion protein was treated with SUMO protease at 4 °C in 50 mM HEPES, pH 7.5, 300 mM KCl, 5% glycerol, 2 mM MgCl_2_, 1 mM Tris(2-carboxyethyl)phosphine (TCEP; Sigma, Cat#C4706). After an overnight treatment, protein was supplemented with 20 mM imidazole and passed through Ni^2+^-NTA resin. The untagged protein was applied to the Superdex 200 increase 10/300 GL column (Cytiva, Cat#28990944) in 50 mM HEPES, pH 7.5, 300 mM KCl, 5% glycerol, 2 mM MgCl_2_, 2 mM DTT. Peak fractions were assessed by SDS–PAGE and Coomassie staining. Purified proteins were assessed for non-specific RNAse activity and stored at −80 °C.

### 4.2. NMPylation Assays

For competition assays with nucleotide analogs, 10 μl reactions containing 0.5 μM nsp12 and 5 μM nsp9 in NMPylation buffer (25 mM HEPES, pH 7.5, 15 mM KCl, 5% glycerol, 1 mM MgCl_2_, 1 mM DTT) were incubated at 37 °C for 5 min. Then, 25 μM GTP (Cytiva, Cat#27202501), 1 μCi [α^32^P]-GTP (PerkinElmer, Waltham, MA, USA, Cat#BLU006H250UC), and different concentrations of competitors were added to start the reaction. We used commercial remdesivir triphosphate (MedChemExpress, Monmouth Junction, NJ, USA, Cat#GS443902), 2′-Azido-2′-deoxyguanosine-5′-triphosphate (Trilink Biotechnologies, San Diego, CA, USA, Cat#N-1063), 2′-Azido-2′-deoxyuridine-5′-triphosphate (Trilink Biotechnologies, Cat#N-1029), and GTP (Cytiva, Cat#27202501). Diphosphates and triphosphates of AT-527 and sofosbuvir were synthesized at Gilead Scientific. For small molecule inhibitors, 10 μL reaction containing 0.5 μM nsp12, 5 μM nsp9 in NMPylation buffer was incubated with inhibitor at indicated concentration at 37 °C for 5 min prior to the addition of 25 μM GTP+[α^32^P]-GTP. Sources and structural analysis of rifapentine and rifabutin were described in [79]; other inhibitors were fidaxomicin (MedChemExpress, Cat# HY-17580), rutin (Sigma, Cat#R5143), tobramycin (Sigma, Cat#T4014), and suramin (Sigma, Cat#S2671). Following 10 min incubation, reactions were stopped with 4× LDS Sample Buffer (Genscript, Piscataway, NJ, USA, Cat#M00676) supplied with 25 mM EDTA.

### 4.3. RNA Extension Assays

An RNA oligonucleotide GA1 (5′ -AAAAGAAAAGACGCGUAGUUUUCUACGCG- 3′) with Cyanine 5.5 at the 5′-end was obtained from Millipore Sigma. Prior to the reaction, the RNA was annealed in 25 mM HEPES, pH 7.5, 50 mM KCl by heating to 75 °C and then gradually cooling to 4 °C. Reactions were carried out at 37 °C. 500 nM nsp12 variant, 1 μM Nsp7, 1.5 μM Nsp8, and inhibitors were incubated in the transcription buffer (25 mM HEPES, pH 7.5, 15 mM KCl, 5% glycerol, 1 mM MgCl_2_, 1 mM DTT) for 3 min at 37 °C. Following the addition of the RNA scaffold (100 nM final) and NTPs (UTP and CTP, 150 μM final), reactions were incubated for 15 min and quenched with an equal volume of 2× stop buffer (8 M Urea, 20 mM EDTA, 1× TBE, 0.2% bromophenol blue).

### 4.4. Electrophoretic Mobility Shift Assays

1 μM holo-RdRp (nsp12:nsp7:nsp8 = 1:2:4) in transcription buffer was incubated with inhibitors at indicated concentrations at 30 °C for 5 min. Then 100 nM GA1 RNA scaffold was added. After 10 min incubation, reactions were mixed with 10X loading buffer (30% glycerol, 0.2% Orange G) and ran on a 3% agarose gel in 0.5× TBE on ice. The gel was visualized by Typhoon FLA9000 (GE Healthcare, Piscataway, NJ, USA).

### 4.5. Cell-Based Viral Infectivity Assays

Vero E6 cells were plated in 48-well plates at 1 × 10^5^ cells/well and allowed to attach overnight. Cells were infected with SARS-CoV-2 WA1 strain at MOI 0.01 for one hour at 37 °C before the inoculum was removed, wells were washed twice with PBS, and increasing concentrations of compounds or DMSO as vehicle control were added to the wells. Infection was allowed to proceed for 24 h before harvesting supernatants. Viral titers were determined by tissue culture infectious dose 50 (TCID_50_) using the Reed and Muench method [85]. 

### 4.6. Sample Analysis

Protein samples were heated for 4 min at 95 °C and separated by electrophoresis in 4–12% SurePAGE gels (Genscript, Cat#M00654). RNA samples were heated for 2.5 min at 95 °C and separated by electrophoresis in denaturing 9% acrylamide (19:1) gels (7 M Urea, 0.5× TBE). The gels were visualized and quantified using Typhoon FLA9000 (GE Healthcare) and Image Quant. All assays were carried out in triplicates. The means and standard deviation (SD) were calculated by Excel (Version 2205 Build 16.0.15225.20172, Microsoft, Redmond, WA, USA).

## Figures and Tables

**Figure 1 molecules-27-03815-f001:**
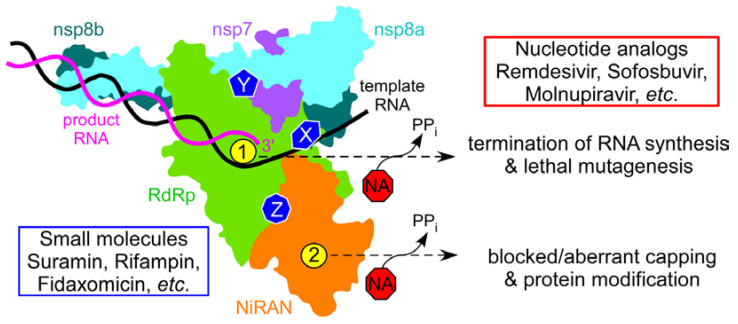
SARS-CoV-2 RdRp holoenzyme is composed of the catalytic nsp12 and accessory nsp7 and nsp8 (present in two copies) subunits. Nsp12 has two active sites, in the RdRp (green) and NiRAN (orange) domains, shown by yellow circles marked as 1 (AS1) and 2 (AS2); the same colors are used throughout the manuscript. The RNA synthesis active site, AS1, is structurally conserved among viral RdRps. NiRAN mediates NMP transfer to viral proteins and to RNA at AS2, which is homologous to AMPylases. Both active sites use diverse NAs as substrates, and the consequences of nucleotide misincorporation vary dependent on the NA. The holoenzyme presents many hypothetical targets for small-molecule inhibitors (X, Y, and Z), which may block RNA binding, interfere with nsp12 contacts to the accessory subunits, or disrupt allosteric communications between the two active sites. Suramin has been shown to bind to two separate sites on nsp12 and inhibit RNA binding. Here, we report that fidaxomicin and rifabutin, natural products in clinical use as antibiotics, inhibit RNA synthesis but not RNA binding; their binding sites and mechanisms of action remain to be identified.

**Figure 2 molecules-27-03815-f002:**
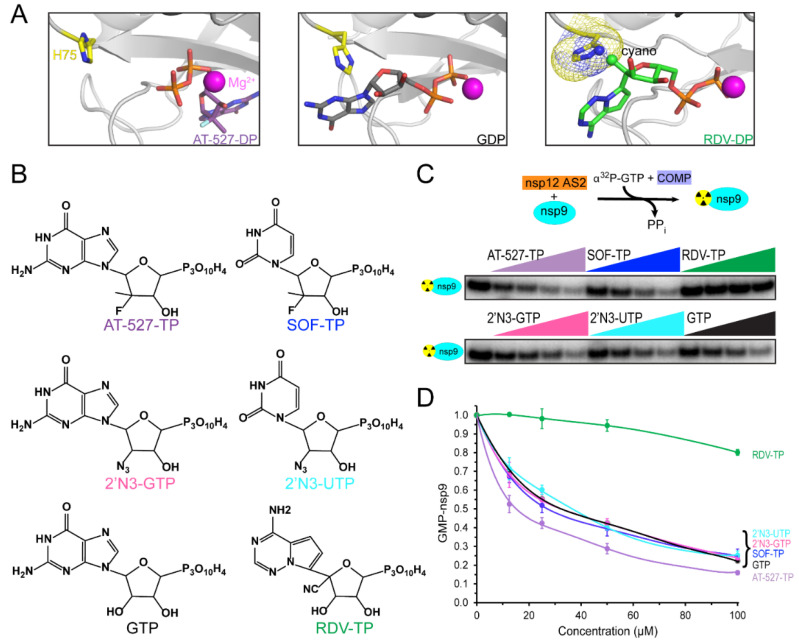
Competition of nucleotide analogs for nsp9 modification. (**A**) Nucleotide binding to the NiRAN active site. The Mg^2+^ ion is shown as a magenta sphere and the His75 residue in the NiRAN active site (AS2) as sticks. Left, AT-527 diphosphate (shown as purple carbon atoms; PDB ID: 7ED5). Center, GDP (black carbon atoms; PDB ID: 7CYQ). Right, RDV-DP (green carbon atoms) was modeled in place of ADP (PDB ID: 6XEZ); the C1′ cyano group of RDV could clash with His75. Structural figures were prepared with PyMOL Molecular Graphics System, version 2.5.2, Schrodinger, LLC (New York, NY, USA). (**B**) Structures of nucleotide analogs tested for competition with [α^32^P]-GTP for transfer to nsp9; these were prepared with ChemDraw 20.1. The color scheme is preserved in all panels. (**C**) Effects of NAs shown in panel B on nsp9 NMPylation by nsp12. Top: the assay schematic; nsp9 and nsp12 were incubated in the presence of [α^32^P]-GTP and indicated analogs. Bottom: [α^32^P]-GMP-nsp9 was detected by protein gel analysis. A representative gel loaded with reactions containing 25 μM GTP and analogs added at 12.5, 25, 50, and 100 μM, from left to right. (**D**). NMPylation efficiency was compared to that observed in the absence of competitors, set at 1, and is shown as mean ± SD (n = 3).

**Figure 3 molecules-27-03815-f003:**
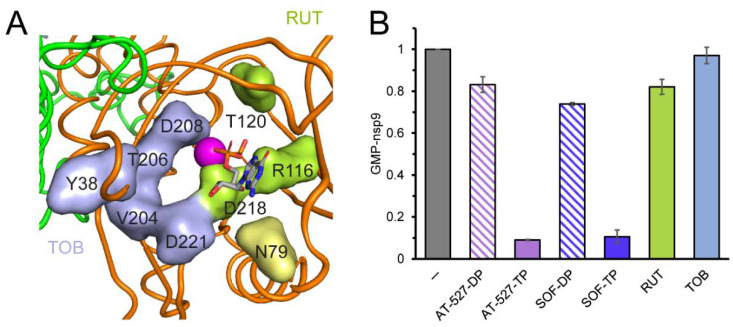
Effects of diphosphates and hypothetical small-molecule inhibitors on NMPylation. (**A**) The NiRAN AS2 (orange) with Mg^2+^ ion shown as a magenta sphere and the GDP as sticks. The nsp12 RdRp domain is in green. Residues identified by in silico docking [30] as contact sites for tobramycin (TOB) are shown in slate, those modeled to bind to rutin (RUT)–in green; Asn79 (yellow) could contact both small molecules. (**B**) Diphosphate (DP) forms of AT-527 and sofosbuvir (SOF) are less effective inhibitors of [α^32^P]-GMP transfer to nsp9 than their triphosphate (TP) forms. TOB has no effect on NMPylation, and RUT has a very small negative effect. All molecules were present at 200 μM in a standard assay shown in Figure 2C. NMPylation efficiency was compared to that observed in the absence of competitors (indicated by a grey bar) and is shown as mean ± SD (n = 3).

**Figure 4 molecules-27-03815-f004:**
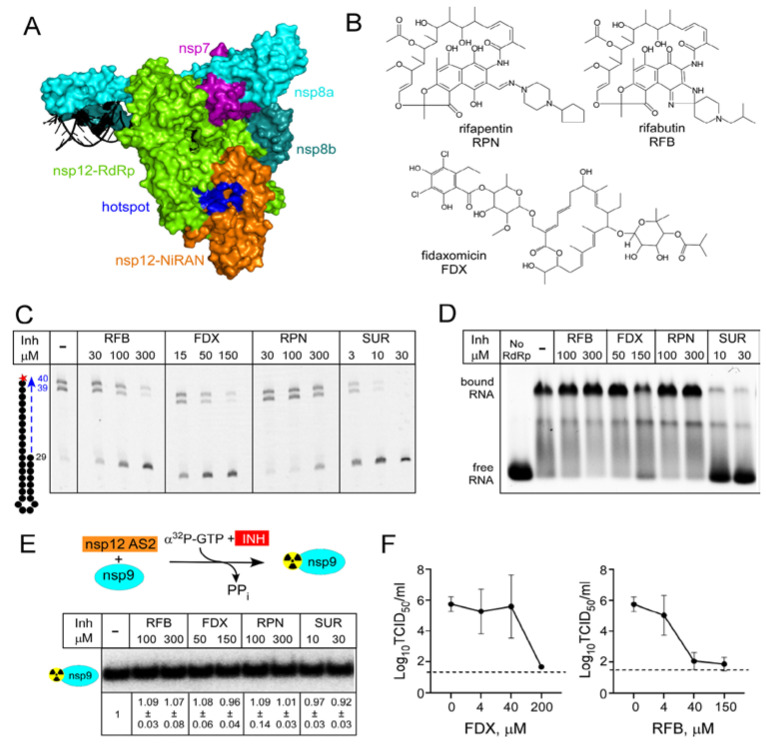
Inhibition of RNA synthesis by SARS-CoV-2 RdRp holoenzyme by small molecules. (**A**) A hypothetical hotspot (blue) at the interface of RdRp and NiRAN (PDB ID: 7CYQ) identified by in silico docking. (**B**) Structures of rifabutin (RFB), rifapentine (RPN), and fidaxomicin (FDX), inhibitors of bacterial RNAP that were docked into the hotspot. (**C**) RNA extension assay in the presence of indicated concentrations of RFB, RPN, FDX, and SUR; RdRp extends the 29-nt RNA hairpin to produce the 39- and 40-mer RNAs. In all assays, DMSO was used as a “no-inhibitor” control. (**D**) Electrophoretic mobility shift assay with the scaffold RNA used in C. The positions of free and RdRp-bound RNAs are indicated. (**E**) RFB, RPN, FDX, and SUR have no effect on nsp9 modification assayed as in Figure 2C. (**F**) Inhibition of SARS-CoV-2 infection. Vero E6 cells were infected with SARS-CoV-2 WA1 strain at MOI 0.01 in the presence of indicated concentrations of compounds. Viral titers were determined 24 h later by TCID_50_ using the Reed and Muench method.

**Figure 5 molecules-27-03815-f005:**
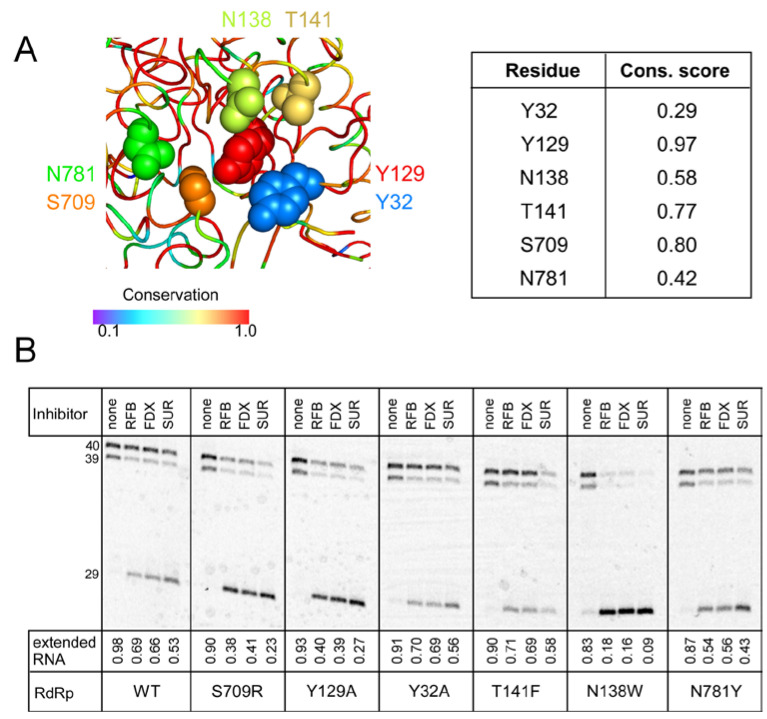
Analysis of the hypothetical hotspot. (**A**) Residues modeled to interact with both FDX and RFB are shown as spheres on the structure of the replicating RdRp (PDB ID: 6YYT), with colors reflecting their conservation scores calculated as described in [23] and shown in the table on the right. (**B**) Effects of substitutions of hotspot residues on RNA synthesis and its inhibition by RFB (100 μM), FDX (30 μM), and SUR (2 μM). A representative gel of 3–4 repeats is shown.

## Data Availability

Data reported here are contained within the article and Appendix A.

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
