# Peer review of "Going Retro, Going Viral: Experiences and Lessons in Drug Discovery from COVID-19"

_molecules, 2022, doi:10.3390/molecules27123815_

Round 1
Reviewer 1 Report
The manuscript entitled: Going retro, going viral: experiences and lessons in drug discovery from COVID-19 is exciting and well-organized. However, I have a few minor concerns listed below:
Overall, the manuscript is too wordy. It should be shortened, especially the discussion and the conclusion section.
Figure #2 Panel C needs some description in the figure legend.
Figure #3 panel B: What does the grey bar represent?
What assay/s was used to determine the TCID50? Adding a description in the method section and Figure #4 legends will be helpful.
The supplementary files (Original images of blots/gels) are not in an accessible format.
Author Response
The manuscript entitled: Going retro, going viral: experiences and lessons in drug discovery from COVID-19 is exciting and well-organized. However, I have a few minor concerns listed below:
Overall, the manuscript is too wordy. It should be shortened, especially the discussion and the conclusion section.
Shortened Discussion and Conclusions, deleted one redundant paragraph from Introduction.
Figure #2 Panel C needs some description in the figure legend.
Added details
Figure #3 panel B: What does the grey bar represent?
The no inhibitor control; revised figure legend.
What assay/s was used to determine the TCID50? Adding a description in the method section and Figure #4 legends will be helpful.
Added details to the Materials and Methods; revised the figure legend
The supplementary files (Original images of blots/gels) are not in an accessible format.
Molecules Instructions for Authors does not specify what formats are allowed. We thus used “original” images, which are gel files as visualized by phosphorimager scanning. This is universal format for those who run gels. We converted these original gels, with annotations, as pdfs and compiled them in one file.
Reviewer 2 Report
Wang et al. describe a series of in vitro experiments to identify efficient inhibitors of the two catalytic sites of the COVID-19 RNA-dependent RNA polymerase (RdRp), starting from a small set of candidate molecules selected with the help of published data and structure-based analyses in silico. The mode of action and in vivo efficacy of a few inhibitors are also investigated. In addition to the experimental results and their immediate implications, the authors provide a lengthy discussion of more general insights (for instance, regarding drug development strategies) that can be gleaned from this study and other work from the COVID-19 era.
The manuscript is well written and makes an engaging read. The experiments are properly designed and convincing.
The only criticism I can think of is that the experimental part (strictly speaking: pages 4-10, which include figures 2-5) is maybe a bit lightweight for a full article. It deals with a rather limited number of inhibitors and binding modes are not investigated in more detail by high-resolution structural methods. In contrast, the 6-page discussion section is very broad and extends well beyond the scope of the experiments. In particular, the text addresses a large body of literature on drug design and repurposing, with a total of 85 references. Hence, the manuscript feels somewhat like a typical “short communication” or “letter” plus a review article or opinion piece (suggested also by the title), merged into one. Whether or not such a mixed format is suitable for Molecules I think is ultimately an editorial decision. My personal view is that the work is of significant interest and raises a number of important points; I would therefore recommend publication of the manuscript regardless of its slightly unusual format.
A few minor issues:
Fig. 1: in the legend to the figure (l. 123-124) as well as in the main text (l. 117, 133, 137,...), the active sites are consistently referred to as “AS1” and “AS2”. In the cartoon, however, the sites appear to be represented by yellow disks marked “1” and “2”, which are not explained in the legend. This is confusing.
Fig. 2: according to the legend to panel C, competitors were present at 25, 50, 100 and 200 μM. In the graph shown in panel D, the concentrations appear to be precisely half those values. Do panels C and D indeed correspond to different experiments, or has there been some kind of mistake?
L. 234: the statement “Structural studies reveal that the ligand-free NiRAN domain is largely disordered” seems to require a reference.
L. 256: “poor substrates for RdRp” should read “poor substrates for AS1”.
L. 295-298, “We note that ... SARS-CoV-2 RdRp activity is very sensitive to the presence of organic solvents”: although this is indeed an important caveat that researchers ought to be aware of, it is not quite clear which observation the authors are trying to explain with this statement. Do they feel that the lack of activity they observe for some compounds might be due to solvent effects? Presumably the appropriate solvent-only control experiments were carried out though? Please clarify.
L. 381, “swords and shields”: I’m not sure this unusual terminology is very helpful, particularly since the words are hardly used at all in the discussion that follows. Moreover, the authors use the expression “double-edged sword” a few times (l. 439, 476), making the “swords and shields” rather confusing.
L. 384, “orthogonal”: the two approaches described here seem to be complementary rather than orthogonal.
L. 417: “against all RNA viruses” should read “against RdRp of all RNA viruses” (the authors do not address the possibility that effective BSAAs might be found against other proteins).
L. 649, “the next one”: presumably the authors mean “the next pandemic” or “emerging diseases”?
Author Response
Wang et al. describe a series of in vitro experiments to identify efficient inhibitors of the two catalytic sites of the COVID-19 RNA-dependent RNA polymerase (RdRp), starting from a small set of candidate molecules selected with the help of published data and structure-based analyses in silico. The mode of action and in vivo efficacy of a few inhibitors are also investigated. In addition to the experimental results and their immediate implications, the authors provide a lengthy discussion of more general insights (for instance, regarding drug development strategies) that can be gleaned from this study and other work from the COVID-19 era.
The manuscript is well written and makes an engaging read. The experiments are properly designed and convincing.
The only criticism I can think of is that the experimental part (strictly speaking: pages 4-10, which include figures 2-5) is maybe a bit lightweight for a full article. It deals with a rather limited number of inhibitors and binding modes are not investigated in more detail by high-resolution structural methods. In contrast, the 6-page discussion section is very broad and extends well beyond the scope of the experiments. In particular, the text addresses a large body of literature on drug design and repurposing, with a total of 85 references. Hence, the manuscript feels somewhat like a typical “short communication” or “letter” plus a review article or opinion piece (suggested also by the title), merged into one. Whether or not such a mixed format is suitable for Molecules I think is ultimately an editorial decision. My personal view is that the work is of significant interest and raises a number of important points; I would therefore recommend publication of the manuscript regardless of its slightly unusual format.
We recognize the issue and have been trying to obtain structural data for a while. However, we think that the data we are reporting is of value to a broader community. We also hope that our views, shaped by decades of studies of molecular mechanisms of bacterial RNA synthesis inhibitors, would be of interest. Hence we chose this “mixed” format.
A few minor issues:
Fig. 1: in the legend to the figure (l. 123-124) as well as in the main text (l. 117, 133, 137,...), the active sites are consistently referred to as “AS1” and “AS2”. In the cartoon, however, the sites appear to be represented by yellow disks marked “1” and “2”, which are not explained in the legend. This is confusing.
Revised the figure legend to clarify
Fig. 2: according to the legend to panel C, competitors were present at 25, 50, 100 and 200 μM. In the graph shown in panel D, the concentrations appear to be precisely half those values. Do panels C and D indeed correspond to different experiments, or has there been some kind of mistake?
Sorry, it was a mistake. The concentrations in the graph are correct, the figure legend was wrong. Now fixed.
L. 234: the statement “Structural studies reveal that the ligand-free NiRAN domain is largely disordered” seems to require a reference.
Added references and more details about NiRAN that have emerged recently.
L. 256: “poor substrates for RdRp” should read “poor substrates for AS1”.
Added AS1 after RdRp to make this even more clear.
L. 295-298, “We note that ... SARS-CoV-2 RdRp activity is very sensitive to the presence of organic solvents”: although this is indeed an important caveat that researchers ought to be aware of, it is not quite clear which observation the authors are trying to explain with this statement. Do they feel that the lack of activity they observe for some compounds might be due to solvent effects? Presumably the appropriate solvent-only control experiments were carried out though? Please clarify.
Bacterial RNAPs easily tolerate 20% DMSO, whereas SARS-CoV-2 RdRp is inhibited at [DMSO] above 2 %. In all our experiments, DMSO-only reactions were always used as controls; we modified Fig. 4 legend and the text to make this clear. While not a problem for experiments reported here, it is a complication.
L. 381, “swords and shields”: I’m not sure this unusual terminology is very helpful, particularly since the words are hardly used at all in the discussion that follows. Moreover, the authors use the expression “double-edged sword” a few times (l. 439, 476), making the “swords and shields” rather confusing.
Removed swords and the associated confusion.
L. 384, “orthogonal”: the two approaches described here seem to be complementary rather than orthogonal.
Deleted orthogonal
L. 417: “against all RNA viruses” should read “against RdRp of all RNA viruses” (the authors do not address the possibility that effective BSAAs might be found against other proteins).
Revised as suggested
L. 649, “the next one”: presumably the authors mean “the next pandemic” or “emerging diseases”?
Added pandemic